# *Synaptotagmin 13* Is Highly Expressed in Estrogen Receptor-Positive Breast Cancer

**Takahiro Ichikawa [1], Masahiro Shibata [1,\*], Takahiro Inaishi [1], Ikumi Soeda [1], Mitsuro Kanda [2], Masamichi Hayashi [2], Yuko Takano [1], Dai Takeuchi [1], Nobuyuki Tsunoda [1], Yasuhiro Kodera [2] and Toyone Kikumori [1]**

1   Department of Breast and Endocrine Surgery, Nagoya University Graduate School of Medicine, 65 Tsurumai-cho, Showa-ku, Nagoya 466-8550, Japan; t.ichikawa@med.nagoya-u.ac.jp (T.I.); t.inaishi@med.nagoya-u.ac.jp (T.I.); iku-soe@med.nagoya-u.ac.jp (I.S.); y.takano@med.nagoya-u.ac.jp (Y.T.); daitakeuchi@med.nagoya-u.ac.jp (D.T.); nobtsun@med.nagoya-u.ac.jp (N.T.); kikumori@med.nagoya-u.ac.jp (T.K.)
2   Department of Gastroenterological Surgery, Nagoya University Graduate School of Medicine, 65 Tsurumai-cho, Showa-ku, Nagoya 466-8550, Japan; m-kanda@med.nagoya-u.ac.jp (M.K.); m-hayashi@med.nagoya-u.ac.jp (M.H.); ykodera@med.nagoya-u.ac.jp (Y.K.)
\*   Correspondence: m-shibata@med.nagoya-u.ac.jp; Tel.: +81-52-744-2251; Fax: +81-52-744-2252

**Abstract:** Background: Accumulating evidence indicates tumor-promoting roles of synaptotagmin 13 (*SYT13*) in several cancers; however, no studies have investigated its expression in breast cancer (BC). This study aimed to clarify the significance of *SYT13* in BC. Methods: *SYT13* mRNA expression levels were evaluated in BC cell lines. Polymerase chain reaction (PCR) array analysis was conducted to determine the correlation between expression levels of *SYT13* and other tumor-associated genes. Then, the association of *SYT13* expression levels in the clinical BC specimens with patients' clinico-pathological factors was evaluated. These findings were subsequently validated using The Cancer Genome Atlas (TCGA) database. Results: Among 13 BC cell lines, estrogen receptor (ER)-positive cells showed higher *SYT13* mRNA levels than ER-negative cells. PCR array analysis revealed positive correlations between *SYT13* and several oncogenes predominantly expressed in ER-positive BC, such as *estrogen receptor 1*, *AKT serine/threonine kinase 1*, and *cyclin-dependent kinases 4*. In 165 patients, ER-positive specimens exhibited higher *SYT13* mRNA expression levels than ER-negative specimens. The TCGA database analysis confirmed that patients with ER-positive BC expressed higher *SYT13* levels than ER-negative patients. Conclusion: This study suggests that *SYT13* is highly expressed in ER-positive BC cells and clinical specimens, and there is a positive association of *SYT13* with the ER signaling pathways.

**Keywords:** breast cancer; *SYT13*; estrogen receptor; progesterone receptor

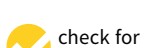

## 1. Introduction

Breast cancer (BC) is the most prevalent malignant tumor among women [1]. Although the prognosis of early BC has improved due to the establishment of neoadjuvant or adjuvant medication therapy, advanced or metastatic BC is still difficult to cure [2,3]. Pharmacotherapies (e.g., chemotherapy, endocrine therapy, and anti-human epidermal growth factor 2 ((HER2)) drugs) have been selected based on the subtypes of tumors according to the expression statuses of hormone receptors (estrogen receptor ((ER)), progesterone receptor ((PgR)) and HER2 because different tumor-growth pathways are activated in various subtypes of tumors [4]. For example, in ER-positive BC, ER signaling has complicated crosstalk with other oncogenic pathways, including the phosphatidylinositol-3-kinase (PI3K)-protein kinase B (AKT) and cyclin-dependent kinases (CDK)4/6 pathways, and the molecules in these pathways are considered novel therapeutic targets [5]. In fact, CDK4/6 inhibitors (e.g., palbociclib, abemaciclib, and ribociclib) combined with anti-estrogen drugs

are clinically efficacious for patients with metastatic ER-positive BC [6,7]. Thus, to further improve the prognosis of patients with BC, it is important to clarify the molecules involved in the signal transduction pathway in each subtype of BC cells.

Synaptotagmins are a group of proteins that play roles in the secretion of neuro-transmitters by synaptic vesicles [8]. The tumor-promoting roles of synaptotagmin 13 (*SYT13*) have been studied in various malignant tumors, including lung adenocarcinoma and colorectal and gastric cancers, and higher *SYT13* expression has been shown to be a poor prognostic factor [9–11]. For example, in gastric cancer, *SYT13* is involved in the progression of peritoneal metastasis and thus is a potential therapeutic target [9]. Although the roles of *SYT13* in promoting tumorigenesis, anti-apoptotic effects, and metastases have been reported in several cancers, no studies have investigated the significance of *SYT13* in BC. Accordingly, this study aimed to investigate whether *SYT13* expression is related to clinicopathological factors, subtypes, and prognoses of patients with BC.

## 2. Materials and Methods

### 2.1. Sample Collection

Among thirteen BC and two non-cancerous breast epithelial cell lines, BT-549, HCC1419, HCC1954, and Hs578T cell lines were purchased from the Japanese Collection of Research Bioresources Cell Bank (Osaka, Japan), and BT-474, MCF-7, and MCF-12A were obtained from the laboratory of Professor David Sidransky at Johns Hopkins University (Baltimore, MD, USA), following the material transfer agreement. Other cell lines were obtained from the American Type Culture Collection (Manassas, VA, USA). These cell lines were cultured in a medium consisting of RPMI 1640 (Sigma-Aldrich, St. Louis, MO, USA) and 10% fetal bovine serum in an atmosphere of 5% $CO_2$ at 37 °C [12].

For clinical specimens, BC patients who were operated on at Nagoya University Hospital from March 2002 to November 2009 and had available postoperative surveillance data spanning more than five years were evaluated in this study. The cancerous specimens were resected at approximately 1.5 mm in diameter and frozen immediately at −80 °C. The resected BC specimens were histologically diagnosed and categorized according to the Union for International Cancer Control (UICC) staging system for BC (8th edition). The perioperative pharmacological treatment of each patient was determined by physician discretion based on the general condition and pathological features.

### 2.2. Quantitative Real-Time Reverse Transcription-Polymerase Chain Reaction (qRT-PCR)

*SYT13* mRNA expression levels were evaluated with qRT-PCR. After total RNA was extracted from cell lines and clinical specimens, cDNA was synthesized [13]. *Glyceraldehyde-3-phosphate de-hydrogenase (GAPDH)* mRNA level was evaluated as a house-keeping gene. The primers specific for *SYT13* and *GAPDH* were as follows: *SYT13*, forward 5-ACCTGGAGAAGGCGAAGC-3 and reverse 5-GTCTGGGAACTTGAGGAGGG-3, which generated a 104-bp product [9]; *GAPDH*, forward 5- GAAGGTGAAGGTCGGAGTC -3, and reverse 5- GAAGATGGTGATGGGATTTC -3, which generated a 226-bp product [12]. A SYBR Green PCR core reagent kit (Applied Biosystems) was used for qRT-PCR with the following cycling conditions: one cycle at 95 °C for 10 min, followed by 40 cycles at 95 °C for 5 s and 60 °C for 60 s. qRT-PCR of each gene was conducted in triplicate. The relative mRNA expression level of *SYT13* was calculated by dividing with the *GAPDH* level [12].

### 2.3. Western Blotting Analysis

Western blotting was performed by the Simple Western technique using the WES instrument (ProteinSimple, San Jose, CA, USA), according to the manufacturer's protocol. Cells were incubated in RIPA lysis buffer, and the lysates were stored at −30 °C. Protein concentration was assessed using a BCA protein assay kit (Thermo Fisher Scientific, Inc., Waltham, MA, USA). Protein samples (5 µg/lane), biotin ladder, primary antibody, secondary antibody, blocking reagent, chemiluminescent substrate, and wash buffer were prepared and dispensed into the assay plate. Then, the assay plate was loaded into the

instrument and the protein was separated into individual capillaries. Protein separation and detection were performed automatically on individual capillaries. The duration of incubation of the primary and secondary antibodies was 30 min at room temperature. Detection was performed by chemiluminescence with luminol (cat. no. 043-311; Protein-Simple) and peroxide (cat. no. 043-379; ProteinSimple) attached to the detection module. Anti-*SYT13* antibody (1:50; cat. no. OAAB02896; Aviva Systems Biology, San Diego, CA, USA) [14], anti-ERα antibody (1:50; cat. no. 8644; Cell Signaling Technology, Danvers, MA, USA), and anti-β-actin antibody (1:50; cat. no. ab6276; Abcam, Cambridge, UK) were used as primary antibodies. Streptavidin HRP (cat. no. 042-414; ProteinSimple) and anti-mouse or anti-rabbit secondary antibodies (anti-mouse, cat. no. 042-205; and anti-rabbit, cat. no. 042-206; ProteinSimple) were selected according to the corresponding primary antibody.

### 2.4. PCR Array Analysis

The RT2 Profiler PCR Array Human for Oncogenes & Tumor Suppressor Genes (Qiagen, Hilden, Germany) was used to investigate the correlation between mRNA expression levels of *SYT13* and 84 cancer-related genes in 13 BC cell lines. The assay was conducted in accordance with the manufacturer's protocol. The relative mRNA expression level of each gene was normalized by the *GAPDH* level.

### 2.5. Public Datasets of BC Cell Lines and Patients

*SYT13* mRNA expression levels in 58 BC cell lines were obtained from Cancer Cell Line Encyclopedia (CCLE) database (https://sites.broadinstitute.org/ccle/). The accessed data was 21 August 2021. Gene expression data and pathological and prognostic characteristics of 681 patients were obtained from The Cancer Genome Atlas (TCGA) database via the cBioPortal for Cancer Genomics (https://www.cbioportal.org/). The accessed date was 26 December 2020.

### 2.6. Statistical Analysis

To compare the continuous valuables of two groups, the Mann–Whitney test was used. On the other hand, to compare multiple groups, ANOVA with Tukey's post hoc test was performed. To analyze the correlation between the expression levels of the two genes, Spearman's rank correlation test was conducted. We used the $\chi^2$ test to evaluate the association between *SYT13* mRNA expression levels and various clinicopathological factors. Prognoses, such as disease-free survival (DFS) and overall survival (OS) rates, were evaluated with the Kaplan–Meier method, following the log-rank test to compare survival curves. JMP 15 software (SAS Institute, Inc., Cary, NC, USA) was exploited for these statistical analyses. $p < 0.05$ was defined as statistical significance.

## 3. Results

### 3.1. SYT13 mRNA Expression Levels in Cell Lines

At first, we evaluated *SYT13* mRNA expression levels in 13 BC cell lines and two non-cancerous mammary cell lines (Figure 1a). The statuses of the conventional biomarkers (ER, PgR, and HER2) are cited from previous studies [15,16]. BC cell lines showed higher *SYT13* mRNA levels than non-cancerous cell lines ($p = 0.024$). ER-positive cell lines showed higher *SYT13* mRNA expression levels than ER-negative ($p = 0.012$) and non-cancerous cell lines ($p = 0.009$). There were no significant differences in *SYT13* mRNA expression between PgR-positive and PgR-negative cells ($p = 0.073$) or between HER2-positive and HER2-negative cells ($p = 0.260$). *SYT13* mRNA expression levels in additional BC cell lines were obtained from the CCLE database for validation (Table S1). ER and PgR statuses in each cell line were referred from previous articles [17–19]. Interestingly, among 58 BC cells whose *SYT13* expression levels were available, ER-positive cells ($n = 17$) displayed significantly higher *SYT13* mRNA levels than ER-negative ones ($n = 41$, $p < 0.001$; Figure 1b) and PgR-positive specimens ($n = 9$) also exhibited higher *SYT13* mRNA expression levels than PgR-negative specimens ($n = 49$, $p = 0.005$; Figure 1b).

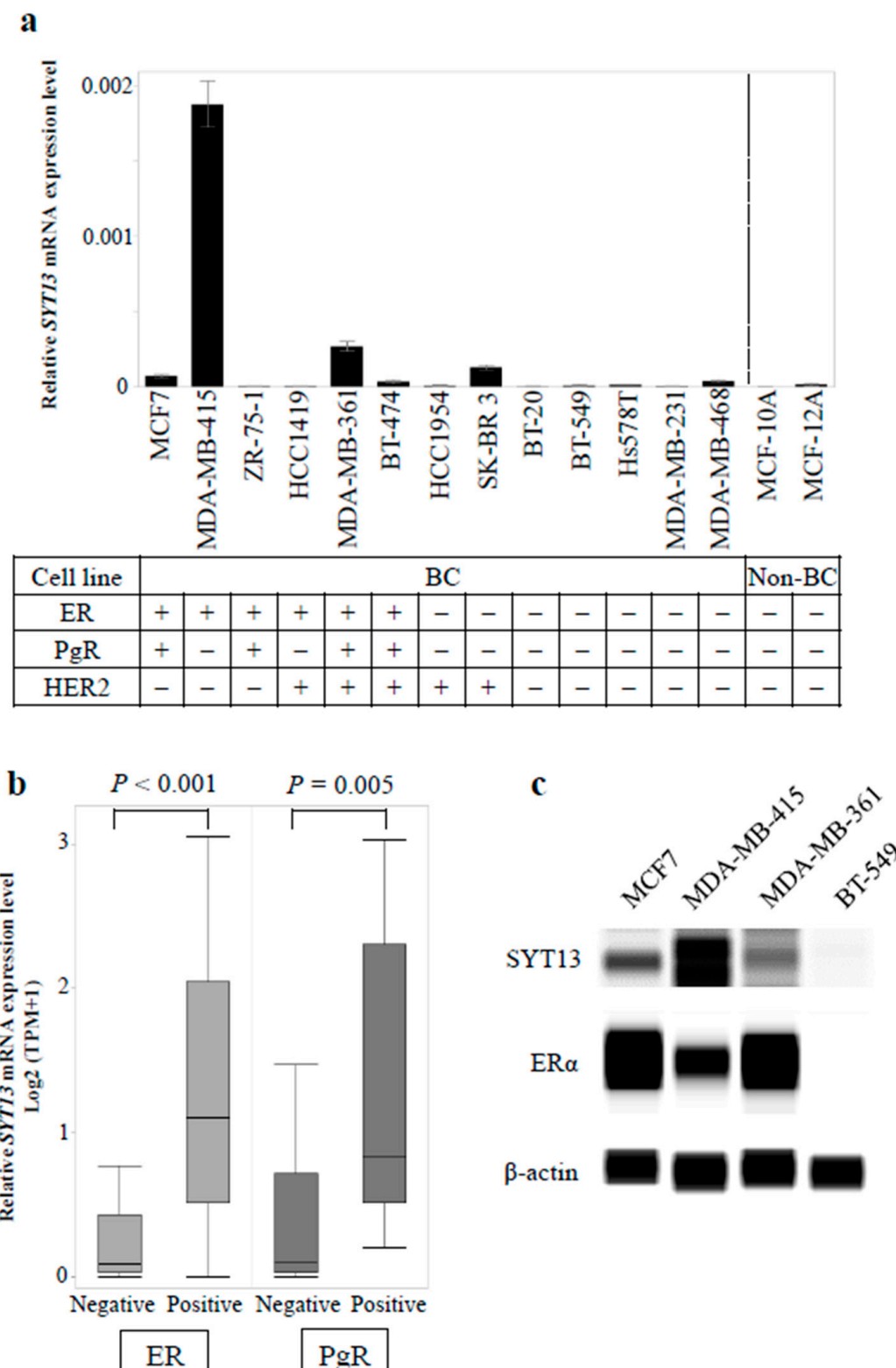

**Figure 1.** (**a**) Expression levels of *SYT13* in breast cell lines. Bar graphs show the relative *SYT13* mRNA levels in 13 BC cell lines and two non-cancerous breast cell lines. ER-positive cell lines showed significantly higher *SYT13* mRNA expression levels than ER-negative cells and non-cancerous cells. (**b**) In the CCLE database, ER-positive and PgR-positive BC cell lines expressed higher *SYT13* mRNA levels, compared with ER-negative and PgR-negative ones, respectively. (**c**) *SYT13* protein expression in the representative BC cells. Among ER-positive cell lines, MDA-MB-415 displayed the highest *SYT13* expression and *SYT13* expressions in MCF7 and MDA-MB-361 cells were modest. BT-549, an ER-negative cell line, hardly expressed *SYT13* or ERα. BC cells, BC cell lines; non-BC, non-cancerous cell lines; TPM, transcripts per million.

Subsequently, *SYT13* protein expression was evaluated in the representative BC cell lines by Western blotting. Among ER-positive cell lines, MDA-MB-415 displayed the highest *SYT13* expression and *SYT13* expressions in MCF7 and MDA-MB-361 cells were modest, which is consistent with the qRT-PCR results (Figure 1c). Alternatively, BT-549, which represented ER-negative cell line, did not express *SYT13* or ERα (Figure 1c).

The correlations between *SYT13* and 84 cancer-related gene expression levels in 13 BC cells were evaluated by PCR array analysis, and it was found that the expression levels of several oncogenes, including *murine double minute 2 (MDM2)*, *AKT serine/threonine kinase 1 (AKT1)*, *B-cell lymphoma 2 (BCL2)*, *estrogen receptor 1 (ESR1)*, and *CDK4*, were positively correlated with those of *SYT13* (Table 1 and Table S2). These genes are known as key genes in the ER signaling pathway that contributes to tumor progression, especially in ER-positive BC [20–22], suggesting involvement of *SYT13* in the ER signaling pathway in BC cells.

**Table 1.** Correlations between mRNA expression levels of *SYT13* and cancer-related genes.

| Gene | Official Full Name | Correlation Coefficient | *p*-Value |
|------|--------------------|-------------------------|-----------|
| *MDM2* | MDM2 proto-oncogene | 0.797 | 0.001 |
| *KRAS* | KRAS proto-oncogene, GTPase | 0.791 | 0.001 |
| *KIT* | KIT proto-oncogene, receptor tyrosine kinase | 0.725 | 0.005 |
| *NFKBIA* | NFKB inhibitor alpha | 0.725 | 0.005 |
| *MYB* | MYB proto-oncogene, transcription factor | 0.709 | 0.007 |
| *AKT1* | AKT serine/threonine kinase 1 | 0.681 | 0.010 |
| *WWOX* | WW domain containing oxidoreductase | 0.681 | 0.010 |
| *XRCC1* | X-ray repair cross complementing 1 | 0.681 | 0.010 |
| *BCL2* | BCL2 apoptosis regulator | 0.670 | 0.012 |
| *TP73* | tumor protein p73 | 0.665 | 0.013 |
| *APC* | APC regulator of WNT signaling pathway | 0.654 | 0.015 |
| *ESR1* | estrogen receptor 1 | 0.654 | 0.015 |
| *BRCA2* | BRCA2 DNA repair associated | 0.648 | 0.017 |
| *BRCA1* | BRCA1 DNA repair associated | 0.643 | 0.018 |
| *RARA* | retinoic acid receptor alpha | 0.637 | 0.019 |
| *MYCN* | MYCN proto-oncogene, bHLH transcription factor | 0.610 | 0.027 |
| *CDK4* | cyclin dependent kinase 4 | 0.599 | 0.031 |
| *NF1* | neurofibromin 1 | 0.593 | 0.033 |
| *CDH1* | cadherin 1 | 0.566 | 0.044 |

*3.2. Association of SYT13 mRNA Expression Levels with Clinicopathological Factors in BC Patients*

In total, 165 female patients with BC were evaluated. The median age was 55 years (27–78 years). Patient demographics are shown in Table 2. If any of ER, PgR, and HER2 was positive, patients were grouped as 'non-triple-negative'. Since eight of the nine patients whose HER2 statuses were unknown showed ER-positivity, these patients were categorized as 'non-triple-negative'.

There were no significant differences of *SYT13* mRNA expression levels between Tis/T1 ($n = 77$) and T2/T3/T4 ($n = 88$; $p = 0.424$), lymph node metastasis-positive ($n = 80$) and negative ($n = 85$; $p = 0.303$), or stage 0/I/II ($n = 131$) and stage III/IV ($n = 34$; $p = 0.732$). Notably, ER-positive specimens ($n = 125$) expressed significantly higher *SYT13* mRNA levels than ER-negative specimens ($n = 40$, $p = 0.005$; Figure 2a), and PgR-positive specimens ($n = 113$) also exhibited higher *SYT13* mRNA expression levels than PgR-negative specimens ($n = 52$, $p = 0.031$; Figure 2a). Moreover, triple-negative patients ($n = 18$) expressed lower *SYT13* than those with non-triple-negative ($n = 146$, $p = 0.046$; Figure 2a). There was no significant difference between HER2-positive ($n = 38$) and negative ($n = 118$) specimens ($p = 0.193$; Figure 2a). In addition, ER-positive/PgR-positive specimens ($n = 113$) expressed higher *SYT13* mRNA levels than ER-negative/PgR-negative specimens ($n = 41$, $p = 0.005$;

Figure 2b), and ER-positive/PgR-negative specimens (*n* = 11) also showed higher *SYT13* mRNA expression levels than ER-negative/PgR-negative specimens (*n* = 41, *p* = 0.024; Figure 2b).

**Table 2.** Clinicopathological characteristics of 165 patients with breast cancer.

| Clinicopathological Parameter | |
|---|---|
| Age, median (range) | 55 (27–78) |
| ≤60 years | 107 (64.8%) |
| >60 years | 58 (35.2%) |
| Histology | |
| DCIS | 7 (4.2%) |
| IDC | 146 (88.6%) |
| ILC | 7 (4.2%) |
| Others | 5 (3.0%) |
| UICC T factor | |
| Tis | 7 (4.2%) |
| T1 | 70 (42.4%) |
| T2 | 74 (44.9%) |
| T3 | 8 (4.9%) |
| T4 | 6 (3.6%) |
| Node status | |
| Negative | 85 (51.5%) |
| Positive | 80 (48.5%) |
| UICC pathological stage | |
| 0 | 7 (4.2%) |
| I | 47 (18.2%) |
| II | 78 (57.3%) |
| III | 32 (22.9%) |
| IV | 1 (0.6%) |
| ER status | |
| Positive | 125 (75.8%) |
| Negative | 40 (24.2%) |
| PgR status | |
| Positive | 113 (68.5%) |
| Negative | 52 (31.5%) |
| HER2 status | |
| Positive | 38 (23.0%) |
| Negative | 118 (71.5%) |
| Unknown | 9 (5.5%) |
| Triple-negative | |
| Yes | 18 (10.9%) |
| No | 146 (88.5%) |
| Unknown | 1 (0.6%) |
| Adjuvant therapy | |
| Endocrine therapy alone | 57 (34.5%) |
| Chemotherapy alone | 30 (18.2%) |
| Endocrine and chemotherapy | 62 (37.6%) |
| None | 16 (9.7%) |

Pathological stages were classified using the UICC staging system for breast cancer (8th edition). DCIS, ductal carcinoma in situ; ER, estrogen receptor; HER2, human epidermal growth factor 2; IDC, invasive ductal carcinoma; ILC, invasive lobular carcinoma; PgR, progesterone receptor; UICC, Union for International Cancer Control.

When patients with *SYT13* expression levels in the highest quartile were allocated to the "high *SYT13* group" (*n* = 42) and the remaining patients were allocated as "others" (*n* = 123), the high *SYT13* group included more ER-positive, PgR-positive, and HER2-negative patients than the "others" group (Table 3). There were no significant differences in T stage (*p* = 0.162), lymph node status (*p* = 0.084), or UICC staging (*p* = 0.095; Table 3). The prognostic analysis did not show significant differences between the high *SYT13* group and the "others" group in DFS rates (5-year DFS rates: high *SYT13* group, 85.1%; others,

81.9%; *p* = 0.752) (Figure 2c) or OS rates (5-year OS rates: high *SYT13* group, 90.1%; others, 89.4%; *p* = 0.220).

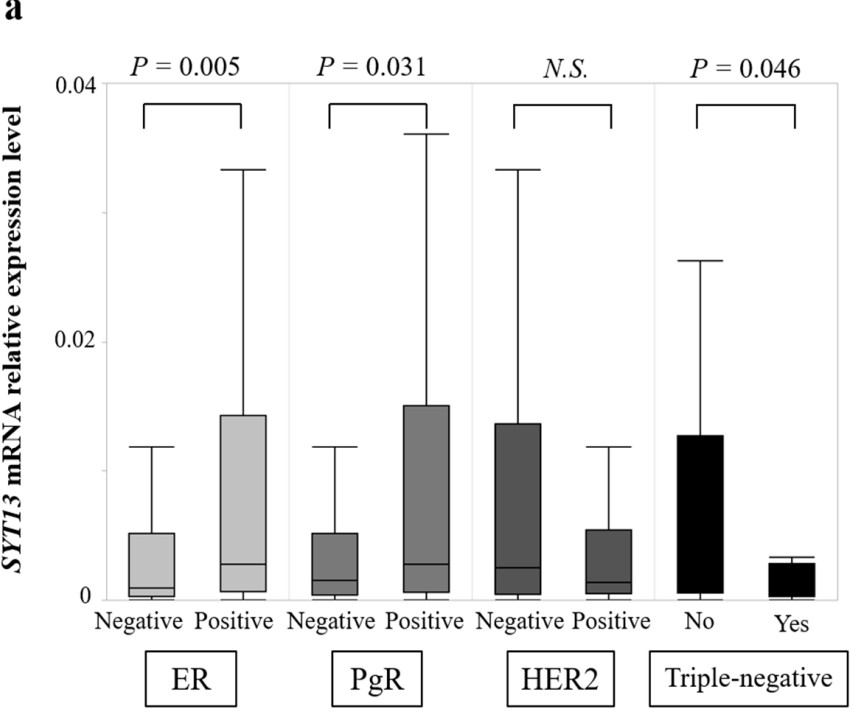

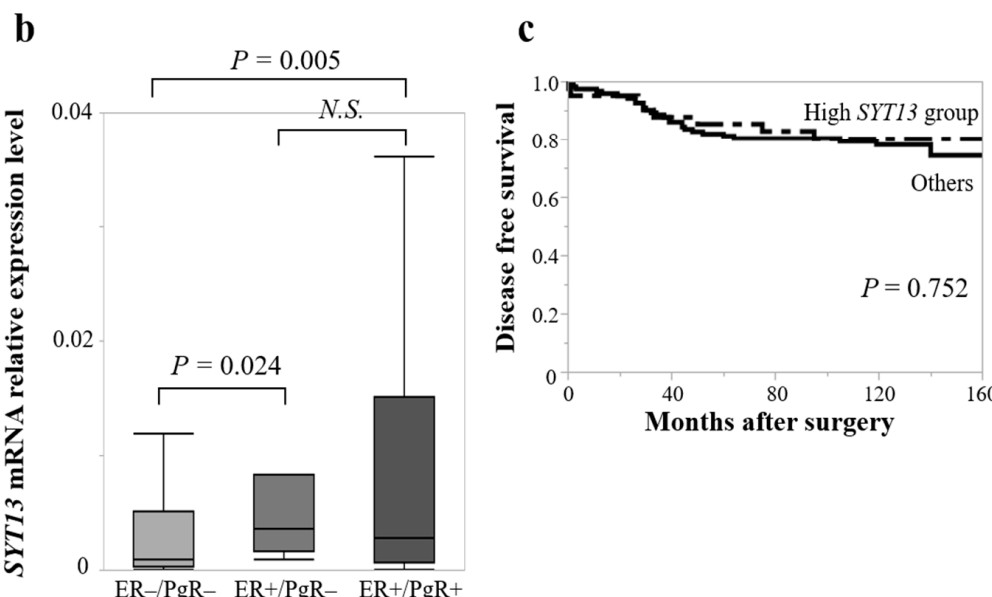

**Figure 2.** (**a**) In the clinical samples, ER-positive specimens exhibited higher *SYT13* mRNA expression levels than ER-negative specimens. PgR-positive specimens also exhibited higher *SYT13* mRNA expression levels than PgR-negative specimens. Triple-negative patients expressed lower *SYT13* than those with non-triple-negative. (**b**) ER-positive/PgR-positive and ER-positive/PgR-negative specimens exhibited higher *SYT13* mRNA expression than ER-negative/PgR-negative specimens. (**c**) There was no significant difference in disease-free survival rate between the high *SYT13* group and the "others" group. N.S., not significant.

**Table 3.** Associations between *SYT13* mRNA expression and clinicopathological characteristics of 165 patients with breast cancer.

| Clinicopathological Parameter | High *SYT13* Group (*n* = 42) | Others (*n* = 123) | *p*-Value * |
|---|---|---|---|
| Age | | | |
| ≤60 year | 28 | 79 | 0.965 |
| >60 year | 15 | 43 | |
| Histology | | | |
| DCIS | 2 | 5 | |
| IDC | 38 | 108 | 0.985 |
| ILC | 3 | 4 | |
| Others | 0 | 5 | |
| UICC T factor | | | |
| Tis/T1 | 24 | 53 | 0.162 |
| T2/T3/T4 | 19 | 69 | |
| Node status | | | |
| Negative | 27 | 58 | 0.084 |
| Positive | 16 | 64 | |
| UICC pathological stage | | | |
| 0/I/II | 38 | 94 | 0.095 |
| III/IV | 5 | 28 | |
| ER status | | | |
| Positive | 38 | 87 | 0.018 * |
| Negative | 5 | 35 | |
| PgR status | | | |
| Positive | 36 | 77 | 0.009 * |
| Negative | 7 | 45 | |
| HER2 status | | | |
| Positive | 4 | 34 | 0.020 * |
| Negative | 33 | 85 | |
| Triple-negative | | | |
| Yes | 3 | 15 | 0.337 |
| No | 39 | 107 | |
| Adjuvant therapy | | | |
| Endocrine therapy alone | 17 | 40 | |
| Chemotherapy alone | 3 | 27 | 0.124 |
| Endocrine and chemotherapy | 18 | 44 | |
| None | 5 | 11 | |

* $\chi^2$ test. DCIS, ductal carcinoma in situ; ER, estrogen receptor; HER2, human epidermal growth factor 2; IDC, invasive ductal carcinoma; ILC, invasive lobular carcinoma; PgR, progesterone receptor; *SYT13*, synaptotagmin 13; Tis, carcinoma in situ; UICC, Union for International Cancer Control.

Because *SYT13* mRNA was preferentially expressed in ER-positive specimens, we evaluated the significance of its expression exclusively in ER-positive patients (*n* = 125). We found no significant differences in *SYT13* mRNA expression between Tis/T1 (*n* = 62) and T2/T3/T4 (*n* = 63; *p* = 0.587), lymph node metastasis-positive (*n* = 63) and negative (*n* = 62; *p* = 0.141), or stage 0/I/II (*n* = 104) and stage III/IV (*n* = 21; *p* = 0.797).

These results from clinical samples showed a positive association between *SYT13* expression levels and ER-positive BC; the finding is consistent with the BC cell line data. However, *SYT13* expression did not affect the prognoses of either all or ER-positive BC patients.

### 3.3. TCGA Database Analysis

To validate the results obtained from our clinical samples, *SYT13* mRNA expression levels in 681 patients with BC were evaluated using the TCGA database. Patient characteristics are summarized in Table 4. *SYT13* mRNA expression levels did not differ between

T1 (*n* = 188) and T2/T3/T4 (*n* = 492; *p* = 0.694), lymph node metastasis-positive (*n* = 354) and negative (*n* = 323; *p* = 0.593), or stage I/II (*n* = 514) and stage III/IV (*n* = 162; *p* = 0.524). Based on hormone receptor status, ER-positive patients (*n* = 532) exhibited higher *SYT13* expression levels than ER-negative patients (*n* = 149, *p* < 0.0001; Figure 3); similarly, PgR-positive patients (*n* = 456) showed higher *SYT13* expression levels than PgR-negative patients (*n* = 225, *p* < 0.0001; Figure 3a). To evaluate patients' prognoses, patients with *SYT13* expression levels in the highest quartile were allocated to the "high *SYT13* group" (*n* = 133), and the remaining patients were allocated as "others" (*n* = 399). There were no significant differences between the high *SYT13* group and the "others" group in DFS rates (5-year DFS rates: high *SYT13* group, 74.6%; others, 83.4%; *p* = 0.479 (Figure 3b). These results are consistent with those from our clinical samples.

**Table 4.** Clinicopathological characteristics of 681 patients in the TCGA database.

| Clinicopathological Parameter | |
|---|---|
| Age, median (range) | 58 (29–90) |
| ≤60 years | 391 (57.4%) |
| >60 years | 290 (42.6%) |
| Sex | |
| Male | 9 (1.3%) |
| Female | 672 (98.7%) |
| Histology | |
| IDC | 494 (72.5%) |
| ILC | 113 (16.6%) |
| Others | 74 (10.9%) |
| UICC T factor | |
| T1 | 188 (27.6%) |
| T2 | 382 (56.1%) |
| T3 | 88 (12.9%) |
| T4 | 22 (3.2%) |
| Unknown | 1 (0.2%) |
| Node status | |
| Negative | 323 (47.4%) |
| Positive | 354 (52.0%) |
| Unknown | 4 (0.6%) |
| UICC pathological stage | |
| I | 124 (18.2%) |
| II | 390 (57.3%) |
| III | 156 (22.9%) |
| IV | 6 (0.9%) |
| Unknown | 5 (0.7%) |
| ER status | |
| Positive | 532 (78.1%) |
| Negative | 149 (21.9%) |
| PgR status | |
| Positive | 456 (67.0%) |
| Negative | 225 (33.0%) |

Pathological stages were classified using the UICC staging system for breast cancer (8th edition). ER, estrogen receptor; IDC, invasive ductal carcinoma; ILC, invasive lobular carcinoma; PgR, progesterone receptor; TCGA, The Cancer Genome Atlas; UICC, Union for International Cancer Control.

Furthermore, no significant differences were found between T1 (*n* = 154) and T2/T3/T4 (*n* = 378; *p* = 0.849), lymph node metastasis-positive (*n* = 237) and negative (*n* = 292; *p* = 0.352), or stage I/II (*n* = 401) and stage III/IV (*n* = 128; *p* = 0.621) in *SYT13* mRNA expression levels in ER-positive patients (*n* = 532); the finding is consistent with the results of our clinical samples.

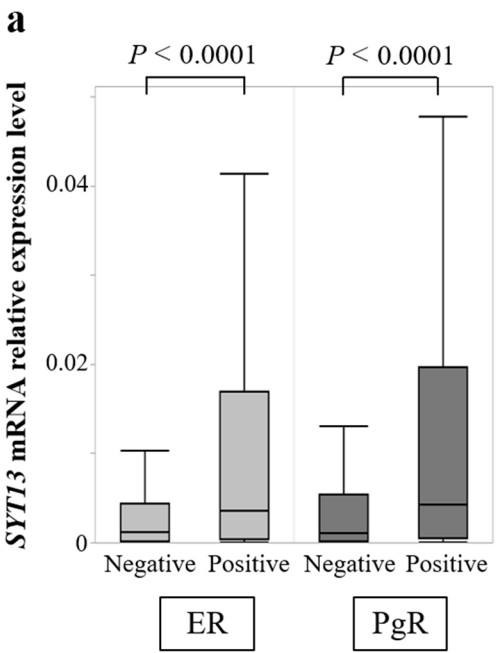

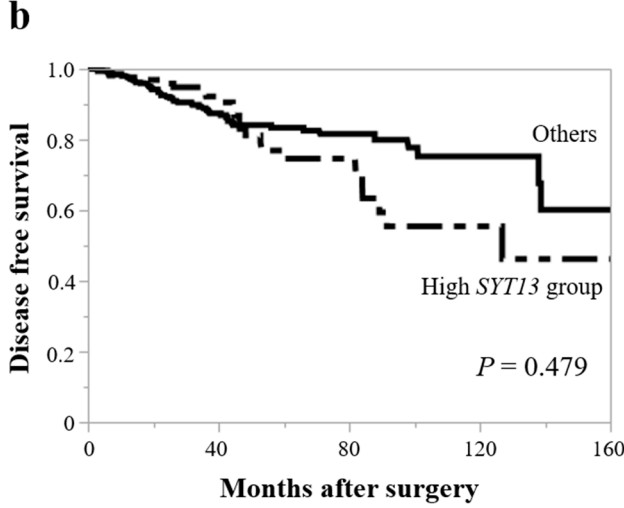

**Figure 3.** TCGA database analysis. (**a**) *SYT13* mRNA expression levels were higher in ER-positive patients than in ER-negative patients. Similarly, PgR-positive patients showed higher *SYT13* expression levels than PgR-negative patients. (**b**) There were no significant differences between the high *SYT13* group and others in disease-free survival rates.

## 4. Discussion

This study demonstrated that *SYT13* was preferentially expressed in hormone receptor-positive BC cell lines and clinical samples based on biochemical analyses and the TCGA database analyses, but there was no significant relationship between its expression and tumor-nodes-metastasis (TNM) staging or prognosis.

*SYT13* is a member of synaptotagmins, a group of proteins that play a role in the secretion of neurotransmitters by synaptic vesicles [8]. The tumor-promoting roles of *SYT13* have been reported in various malignant tumors, such as lung adenocarcinoma and colorectal and gastric cancers [9–11]. In lung adenocarcinoma, *SYT13* contributes to cellular proliferation, clonality, and anti-apoptotic effects [11], and in colorectal cancer, silencing *SYT13* inhibits tumor growth in mouse models [10]. Our group previously found that intraperitoneal injection of *SYT13* siRNA inhibited the peritoneal dissemination of gastric cancer in mouse models, and that *SYT13* expression was an independent risk factor for

peritoneal recurrence in patients with gastric cancer [9,14]. Despite these previous findings, the significance of *SYT13* in BC remains unclear.

Initially, we had anticipated that patients with advanced BC would have higher *SYT13* mRNA expression levels, which indicates a poor prognosis, as demonstrated in other cancer types. However, unlike other malignancies, we detected no significant associations in *SYT13* expression levels among TNM stages or prognosis in our cohort or the TCGA database. Regarding the significance of *SYT13* expression, it is necessary to take into account the biological differences between breast and other cancers that have been reported previously. We subsequently investigated whether *SYT13* expression levels were related to conventional biomarkers, such as ER, PgR, and HER2. We found that ER-positive BC cells and clinical specimens had higher *SYT13* mRNA expression levels than ER-negative cells and clinical specimens, respectively. In addition, *SYT13* protein expression was consistent with mRNA expression levels. The CCLE database, which includes a larger amount of cell lines' data, validated our results. In our clinical samples, *SYT13* mRNA expression levels were also higher in PgR-positive and non-triple-negative patients. In the TCGA database, a larger and more comprehensive cohort dataset, ER-positive and PgR-positive patients had significantly higher *SYT13* expression levels. These results consistently suggest that *SYT13* expression correlates with the positivity of hormone receptors but does not affect the stage of progression in BC.

In ER-positive BC, ER signaling is mainly activated to promote the transcription of molecules involved in cell proliferation, thereby resulting in tumor progression [5]. Importantly, ER signaling is known to crosstalk with other oncogenic pathways, such as the PI3K-AKT and CDK4/6 pathways [5]. Consistently, our PCR array analysis demonstrated positive correlations between expression levels of *SYT13* and *ESR1*, *AKT1*, *MDM2*, *BCL2*, and *CDK4*. To date, no reports have described the relationship between *SYT13* and these genes. The positive correlation between *SYT13* and *ESR1*, a transcription factor, supports our finding that *SYT13* is highly expressed in ER-positive patients. *AKT1* is one of the downstream targets of *ESR1* and it is also overexpressed in ER-positive BC [21]. In small bowel neuroendocrine tumors, *SYT13* has been reported to be involved in metastasis by interacting with the AKT pathway [23]. Both *MDM2* and *BCL2*, downstream of *AKT1*, are overexpressed in ER-positive BC and have been considered potential therapeutic targets for BC [22]. CDK4, which facilitates phosphorylation of Rb, accelerates the cell cycle [20]. Recently, several studies have shown the importance of the CDK4/6 and PI3K/AKT/mTOR pathways in ER-positive/HER2-negative BC [24]. CDK4/6 inhibitors with hormone therapy are globally used as a first-line treatment for metastatic ER-positive BC [6,7]. It would be interesting to further investigate the association between the efficacies of these inhibitors and *SYT13* expression in patients with metastatic ER-positive BC. The correlations between expression levels of *SYT13* and these genes suggest that *SYT13* exists in ER signaling pathways; the hypothesis is supported by our results of clinical samples. However, *SYT13* expression was not associated with tumor staging or prognosis in either all or ER-positive BC patients. These results suggest that although *SYT13* exists in the ER-related pathways, it is not directly involved in promoting BC. In other words, *SYT13* is considered to play a subordinate role in ER-positive BC.

One limitation should be mentioned in this study. Mechanistic experiments were lacking. Although PCR array analysis revealed the association between *SYT13* and other cancer-related genes that are mainly expressed in ER-positive BC, these observational results are not capable of determining the crosstalk between these genes.

## 5. Conclusions

This study found increased expression of *SYT13* in ER-positive BC. The present findings suggest that *SYT13* has the potential to bridge oncogenic pathways with the ER signaling pathway in BC, which would contribute to clarifying the full picture of the ER signaling pathway.

**Supplementary Materials:** The following are available online at https://www.mdpi.com/article/10.3390/curroncol28050346/s1, Table S1. *SYT13* mRNA expression levels of 58 BC cell lines in the CCLE database. Table S2. Correlations between mRNA expression levels of *SYT13* and 84 cancer-related genes.

**Author Contributions:** Conceptualization: M.S. and M.K.; experiments: T.I. (Takahiro Ichikawa), M.S., T.I. (Takahiro Inaishi), and I.S.; data analysis: T.I. (Takahiro Ichikawa); resources (cell lines): M.H.; data and sample collection: T.I. (Takahiro Ichikawa), M.S., T.I. (Takahiro Inaishi), I.S., Y.T., D.T., N.T., and T.K.; writing the manuscript: T.I. (Takahiro Ichikawa) and M.S.; supervision: Y.K. and T.K. The final manuscript has been approved by all authors. All authors have read and agreed to the published version of the manuscript.

**Funding:** This research received no external funding.

**Institutional Review Board Statement:** This study was conducted in accordance with the guidelines of the Declaration of Helsinki and approved by the Institutional Review Board of Nagoya University Graduate School of Medicine (reference number: 2019-0028).

**Informed Consent Statement:** Participants in this study provided written informed consent for the use of their clinical samples and data.

**Data Availability Statement:** The data used in this study can be obtained from the corresponding author upon request.

**Acknowledgments:** We are grateful to David Sidransky, the director of the Otolaryngology Department of Johns Hopkins University School of Medicine (Baltimore, MD, USA), for providing the BT-474, MCF-7, and MCF-12A cell lines.

**Conflicts of Interest:** The authors declare no conflict of interest.

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
