# Peer review of "Synaptotagmin 13 Is Highly Expressed in Estrogen Receptor-Positive Breast Cancer"

_curroncol, doi:10.3390/curroncol28050346_

Round 1

Reviewer 1 Report

The comments raised by the Reviewer 3 were well taken and the manuscript was revised appropriately.  Thus, the manuscript will be acceptable for publication in current form.

Author Response

We do appreciate the approval comments from the Reviewer. Thanks to the Reviewer's valuable suggestions, our results has been strengthened by evaluating the CCLE database and protein expressions in the cell lines.

Reviewer 2 Report

An original study exploring the role of synaptotagmin 13 expression in breast cancer, showing that this molecule is highly expressed in this tumor and that has a positive association with the tumor signaling pathway. The paper has already been reviewed, so only minor corrections are necessary:

Conclusions should be broadened better describing the future developments following this study's results.

page 1 line 34-38 "Pharmacotherapies (e.g., chemotherapy, endocrine therapy, and anti-human epidermal growth factor 2 (HER2) drugs) have been selected based on the subtypes of tumors according to the expression statuses of hormone receptors (estrogen receptor (ER), progesterone receptor (PgR)) and HER2 because different tumor-growth pathways are activated in various subtypes of tumors." this paragraph needs a reference, such as doi: 10.1007/s40264-021-01071-1.

Author Response

We deeply appreciate the Reviewer for re-reviewing the manuscript promptly and giving us further positive suggestions.

Conclusions should be broadened better describing the future developments following this study's results.

Reply) According to the Reviewer's positive suggestions, we added the significance of our results in the Conclusions. "The present findings suggest that SYT13 has the potential to bridge oncogenic pathways with the ER signaling pathway in BC, which would contribute to clarifying the full picture of the ER signaling pathway (lines 328-329)".

page 1 line 34-38 "Pharmacotherapies (e.g., chemotherapy, endocrine therapy, and anti-human epidermal growth factor 2 (HER2) drugs) have been selected based on the subtypes of tumors according to the expression statuses of hormone receptors (estrogen receptor (ER), progesterone receptor (PgR)) and HER2 because different tumor-growth pathways are activated in various subtypes of tumors." this paragraph needs a reference, such as doi: 10.1007/s40264-021-01071-1.

Reply) We thank the Reviewer for kindly showing the paper. We cited the indicated literature (ref. no. 4: line 38).

This manuscript is a resubmission of an earlier submission. The following is a list of the peer review reports and author responses from that submission.

Round 1

Reviewer 1 Report

The manuscript describes the role of the synaptotagmins in breast cancer. Although the synaptotagmin 13 (SYT13) has been related with progression in other cancers, they do not found any correlation with progression in breast cancer.

It is very interesting their correlation between ER-positive specimens and several oncogenes related with the pathway of ER signaling, such MDM2, AKT, ESR1 and specially with CDK4 that is a very important target of the estradiol in breast cancer.

We must consider SYT13 as an important biomarker of luminal breast cancer due to its high correlation with other biomarkers of the luminal phenotype. They found this correlation in three studies, with cell lines, tumor samples, and analysis from the TCGA database; Therefore, these results are very strong to indicate a relevant role in the luminal phenotype of breast cancer.

The fact that a prognostic value has not been found because survival benefit is not associated does not diminish its important role in the biology of breast cancer. Although if this association has been found in other tumors, it is necessary to take into account the difference between breast cancer and other types of tumors, both due to their own biology and the efficacy of the treatments.

I think it would be very interesting to analyze the role of SYT13  in the metastatic stage and in patients with luminal phenotype. It is possible that in this context a prognostic value or even a predictive value of efficacy to combined treatments of hormonal therapy with cyclin inhibitors can be found, since it is associated with the expression of CDK4 / 6 and other components that exist in the hormonal signaling cascade.

Reviewer 2 Report

The authors of this manuscript have investigated the expression of Synaptotagmin 13 (SYT 13) in breast cancer. This is interesting as SYT 13 has been studied in other organs and a possible tumour-promoting role has been explored, particularly in gastric/colorectal cancer. The data presented is interesting and show an association between SYT13 expression and some histo-pathological features of breast cancer. However, the manuscript is more a series of observation and there is no attempt at establishing any correlation between SYT13 and any neoplastic mechanism (as acknowledge at the end of the discussion). As such, this manuscript looks more like the first part of a longer article where such mechanisms are explored. The manuscript would be more interesting to the field if such mechanistic data were added.

The data on cell lines only shows that 2 out of 6 lines (MDA-MB-415 and -361) have higher expression on SYT13 while the others are similar to the ER-neg lines. As the experimental details are not explained in details, one can only assume that group of cell lines (say ER+ v ER-, or PgR+ v PgR-) are compared. Is this method valid considering the high value of MDA-MB-415? Eye balling the plot, one could say that if you ignore MDA-MB-415 in the HER2+ v HER2- comparison, there could be a significant difference (possibly also with PgR comparison). The authors should have a look at this. In any case, use of cell lines for such comparison is a fraught enterprise as cell lines have evolved and are very different. Maybe a description of the plot might suffice here.

The correlation between SYT13 and 84 cancer genes is similarly flawed if it uses pooling of cell lines. It might be more informative to look at each individual cell lines (MDA-MB-415, MDA-MB-361, SK-BR3 and possibly MCF7) and see what gene/pathways are associated (I use associated here as correlation is a strong statement and must be proved experiment).  That would be more informative and could lead to some experiments where knock-down of SYT13 should be used to explore effects on ER signaling genes. The authors have in their hand a nice set of cell lines where knock-down/knock-in experiments could be pursued to demonstrate the role of SYT13 in neoplasia.

The only result from the clinical analysis, as well as the TCGA exploration, is an increase SYT13 with ER+ BC. While interesting, this is not a ground-breaking finding. This increase does not lead to a worse outcome in patient. This should really be explored further, possibly along the lines of reference 9. As it stand, this increase could just be a by-stander effect of ER+ BC.

Reviewer 3 Report

The authors measured levels of Synaptotagmin 13 (STY13) transcripts in 13 breast cancer cell lines and patient-derived BC specimens.  They demonstrate that the STY13 mRNA levels are higher in ER-positive cells and patient derived ER-positive BC specimens than ER negative counterparts.  In addition, on the basis of the results obtained by PCR array analysis, the high levels of STY13 transcript is correlated with the expression of several oncogenes such as estrogen receptor 1, AAKT serine/threonine kinase 1 and cyclin-dependent kinase 4 genes.  However, they did not find any significant correlation between the levels of SYT13 mRNA and the stages of BC progression and metastasis in the specimens derived from their BC patients and the data published in TCGA.  STY13 is suggested to promote tumor progression in colorectal cancer (Int J Mol Med 2020, 45:  234-244), however, this is not the case in BC.  Since little is known on biological functions of Synaptotagmins in non-neuronal cells, especially in tumors, the presented work is still potentially interesting after making some revisions in the manuscript.  In particular,

  1. The authors state that there is a good correlation of the STY13 mRNA levels and ER positivity in 13 BC cell lines.  In reality, there is a way to search on the issue utilizing the public data available in the Broad Institute Cancer Cell Line Encyclopedia (CCLE).  In this site, there are deposited data on 19 BC cell lines, 6 of them are overlapped with those used by the authors.   Except for a couple of cell lines (HCC1419 and  HCC38), I would say that there is a good positive correlation between the STY13 mRNA levels and ER positivity.  This kind of search must be performed also on the genes that are identified by PCR array analysis using CCLE database.  Then, the results of the analysis must be included either in the Results or in the Discussion section.
  2. Levels of transcript do not always correspond to those of the respective protein.  If good antibodies are available, it is strongly suggested to perform WB analysis of the STY13, ER and the down-stream target proteins expressed in 2-3 BC cell lines to strengthen a positive association of SYT13 with the ER signaling pathways, as is suggested by the authors.